# The Perceptions of People with Dementia and Key Stakeholders Regarding the Use and Impact of the Social Robot MARIO

**DOI:** 10.3390/ijerph17228621

**Published:** 2020-11-20

**Authors:** Dympna Casey, Eva Barrett, Tanja Kovacic, Daniele Sancarlo, Francesco Ricciardi, Kathy Murphy, Adamantios Koumpis, Adam Santorelli, Niamh Gallagher, Sally Whelan

**Affiliations:** 1School of Nursing and Midwifery, Aras Moyola, NUI, Galway, Ireland; Kathy.murphy@nuigalway.ie (K.M.); ngallagher@nuigalway.ie (N.G.); s.whelan7@nuigalway.ie (S.W.); 2College of Engineering and Science, Alice Perry Building, NUI, Galway, Ireland; eva.e.barrett@nuigalway.ie; 3UNESCO Child and Family Research Centre, School of Political Science and Sociology, NUI, Galway, Ireland; tanja.kovacic@nuigalway.ie; 4Sistemi Informativi, Innovazione e Ricerca, IRCCS Casa Sollievo della Sofferenza. Viale Cappuccini, 1 71013 San Giovanni Rotondo FG, Italy; d.sancarlo@operapadrepio.it (D.S.); f.ricciardi@operapadrepio.it (F.R.); 5Institut Digital Enabling, Berner Fachhochschule, CH-3012 Bern, Switzerland; adamantios.koumpis@gmail.com; 6Faculty of Engineering, Macdonald Engineering Building, 817 Sherbrooke Street West, Room 382 Montreal, Montreal, QC H3A 0C3, Canada; adam.santorelli@mail.mcgill.ca

**Keywords:** dementia, Alzheimer’s, older adults, social robots, companion robots, MARIO, qualitative research, quality of care, long-term care

## Abstract

People with dementia often experience loneliness and social isolation. This can result in increased cognitive decline which, in turn, has a negative impact on quality of life. This paper explores the use of the social robot, MARIO, with older people living with dementia as a way of addressing these issues. A descriptive qualitative study was conducted to explore the perceptions and experiences of the use and impact of MARIO. The research took place in the UK, Italy and Ireland. Semi-structured interviews were held in each location with people with dementia (*n* = 38), relatives/carers (*n* = 28), formal carers (*n* = 28) and managers (*n* = 13). The data was analyzed using qualitative content analysis. The findings revealed that despite challenges in relation to voice recognition and the practicalities of conducting research involving robots in real-life settings, most participants were positive about MARIO. Through the robot’s user-led design and personalized applications, MARIO provided a point of interest, social activities, and cognitive engagement increased. However, some formal carers and managers voiced concern that robots might replace care staff.

## 1. Introduction

Dementia is a progressive neurocognitive disorder that has a profound effect on a person’s personality, memory, social skills, ability to communicate and make decisions as well as on mood and emotional reactions [1,2,3]. Currently there are 50 million people worldwide with this condition, however as the estimated number of people over 60 increases to over two billion people by 2050 [4], and because the incidence of dementia is correlated with increased longevity, it is projected that the number of people with dementia will increase to 82 million by 2030 and 152 million by 2050 [5]. In Europe, it is estimated that the figure will reach 14.3 million by 2050 [6]. Therefore, it is not surprising that dementia is one of the greatest societal and economic challenges associated with ageing in the 21st century [7,8,9,10]. It is imperative, therefore, that strategies are identified to support people with dementia and their families to live well with dementia.

Living well with dementia requires the implementation of interventions that can impact positively on the person’s quality of life. Many people with dementia may live meaningful lives and retain abilities if a supportive psychosocial environment exists. Spector and Orrell [11] suggest that there are protective/destructive psychosocial factors at play and that social engagement and sustained connectedness are crucial to improving the outcomes for people with dementia. Increasingly the potential of social robots to enhance engagement for people with dementia is recognised as a means of combating loneliness, social isolation and boredom [12,13,14]. This paper presents the perceptions and experiences of people with dementia and key stakeholders as regards the use of the social robot, MARIO, deployed with people with dementia in three different countries and clinical contexts.

Social participation is a critical contributing factor to successful and healthy aging. Indeed, high levels of social participation have been found to be associated with less cognitive impairment and depression, irrespective of physical frailty [15] However, dementia can lead to reduced social engagement, isolation, and loneliness [13,16,17]. Loneliness and social isolation are recognised as major public health issues associated with higher all-cause mortality rates [18,19]. The risk to health due to social isolation has been equated with the risk associated with cigarette smoking, hypertension and obesity [20]. In the UK, over a third of people with dementia reported feeling lonely and had difficulties maintaining social relationships [21]. In the context of long-term care many studies found that residents spend most of their time socially unconnected and not engaged in any meaningful activity [13,22,23,24,25,26,27,28,29]. Such persistent and continued lack of stimulation and social interaction exacerbates further the lethargy, boredom, depression, and loneliness that are often manifest in the progression of dementia [30,31]. Engaging activities and identifying ways of occupying time meaningfully is an essential part of quality of care. Social engagement can enhance the well-being of people with dementia by maintaining their self-esteem and social connectedness as well as providing a purpose for day-to-day living [32,33,34,35]. Social robots are increasingly seen as having the potential to provide such meaningful activities [14] and therefore have a part to play in the overall quality of care.

Social, or companion robots, are defined as robots that have the capability of interacting with people in a socially acceptable way [36]. While these terms are used interchangeably in the literature the term social robots will be used throughout this paper. The development of social robots for the psychosocial wellbeing of people with dementia is a young discipline and a recent area of research. It started with the development of animal shaped zoomorphic robots that built on the success of animal therapy in dementia care. Zoomorphic robots can positively impact the emotions and communication of people with dementia. PARO, which is designed to appear as a baby harp seal has been most widely implemented into care practice [37]. To date there have been at least twenty three EU funded projects that have conducted research into a wide variety of robots. These include the MARIO project (www.mario-project.eu). Currently robots have limited capacity to read human emotions and current development aims to increase their ability to communicate in a more humanlike way [38]. Several studies describe the important role that social robots can play in dementia care by providing companionship and opportunities for people with dementia to engage in meaningful activities [13,39,40,41,42] resulting in improved social engagement [43,44,45,46]. Research has also found that people with dementia are generally positive toward and accepting of social robots [47,48,49,50].

Social robots have been found to have positive effects by reducing negative emotions and behavioural symptoms, improving social engagement, and promoting positive mood and quality of care experience [51]. Additionally, patients who use social robots in a patient-centred manner are perceived as having higher emotional intelligence themselves and can affect caregivers to form more positive impressions of the person that the robot cares for [52]. These findings demonstrate that social robots also have the potential to enhance human-human relationships in the healthcare context. 

Factors that influence the acceptability of social robots include having humanlike facial features, being an embodied presence and having social capabilities [17,53,54] being able to deliver specific personalised activities that meet the needs of the individual end user [17,54] and having reliable technology [55]. The perceptions of significant others, such as relatives or carers are also important in determining the acceptability of social robots [54]. Having positive perceptions toward the use of social robots as a means of communication and providing social engagement for the person with dementia is identified as important [56,57]. However, most studies that examined the use of social robots with people with dementia have been conducted over relatively short testing periods [48,49,57,58] and were conducted in the participant’s home or simulated home set-up in a laboratory rather than in the real world of practice [47,48,59,60].

## 2. Materials and Methods

### 2.1. The MARIO Robot

A multidisciplinary trans-European consortium of researchers, clinical practitioners from community, hospital and residential care settings, ICT specialists and industrial partners with expertise in robotics were assembled with the aim of developing a social robot. In total, the MARIO consortium brought together the skills and expertise of 10 partners from six countries. MARIO is a social robot whose functions aim to support the psychosocial wellbeing of people with dementia, through supplementing the care provided by human carers. MARIO has no functional capacity to address a person’s physical needs. A user-led design process involving people with dementia and other relevant stakeholders was used utilised. This resulted in a 1.5-metre-tall white robot with large animated eyes that moved on wheels and could be activated by voice or touchscreen (Figure 1). An iterative design process was used whereby the applications were developed, and refined based on user preferences, testing and feedback [40,61]. This led to the development of several bespoke applications (Table 1) tailored to the specific needs of each person with dementia. MARIO was deployed in three pilot sites, in different health care contexts a purpose-built long-term care setting (Ireland), a geriatric unit in a hospital (Italy), and a community setting (UK). A MARIO robot arrived in each pilot site equipped with the ability to map out a given location and then subsequently autonomously navigate around the dementia care setting. However, because the clinical practice environment constantly changed, necessitating remapping each time, autonomous navigation was not possible. MARIO therefore was not fully autonomous during this research and interaction sessions were supervised requiring the presence of a researcher to guide navigation and provide assistance as required. However during the final evaluation stage of the research the researcher supervised at a distance to give MARIO as much autonomy as possible.

### 2.2. Ethics

All participants, including people with dementia gave their informed consent for inclusion before they participated in the study and confidentiality was maintained. Suitably qualified health professionals or psychologists, at each pilot site, ensured that participants had capacity to consent following procedures that conformed to national laws, regulations, and best practice in dementia research. Process consent was utilised, in that, consent was sought, not just for involvement in the overall research but, consent was checked again for each interaction with MARIO. The study was guided by experts in ethics who developed and implemented an ethical framework, and the study was conducted in accordance with the Declaration of Helsinki, and the study was approved by Research Ethics Committees in Ireland (REC, NUI, Galway) UK, (REC, Stockport Metropolitan Council) and Italy (REC:Casa Sollievo della Sofferenza) In addition, it was recognised that there was a need for careful management of the disengagement process between MARIO and the person with dementia, particularly for those who had spent longer times with MARIO. Issues connected with the disengagement process were identified as potential ethical challenges and a disengagement plan was utilised.

### 2.3. Research Phases

There were three phases in the research. Phase one and two focused on the acceptability of MARIO and the development of MARIO applications. A description of the five applications (app.) is given in order to contextualise findings. (1) The My Memories app. was designed to facilitate reminiscence, and drew on the preserved memories of the PWD. The researcher gathered, often with the help of family, carers and friends, pictures of relevance to the interests and life of each individual PWD and then uploaded these to the MARIO platform. Mario utilised these pictures to stimulate conversation with the PWD using the pictures as prompts. (2) The MY Music app. enabled the PWD to select what music they listened to, when they listened to it, and switch music if they wished. The researcher created a file containing personalized music preferences and these were uploaded to the MARIO platform. The application built knowledge of selections over time and choices were refined based on usage. (3) The My News app. was linked to news feeds from the web. It allowed the PWD to access news headlines or follow personalised interests, like sports, politics, community events. MARIO could read news items of interest to the PWD or display it in written form on the monitor for the PWD to read. The purpose of the application was to keep the PWD briefed and connected. (4) The My Calendar app. reminded the PWD of events like birthdays, anniversaries, visits from others, appointments, community activities, the app. was personalised to each PWD, and facilitated active participation in community and family events. The My Games app. included a range of games which were personalised to the PWD. Games like chess, drawing, solitaire, puzzles, bingo, tennis, painting could be selected as preferences. The aim of this app. was to stimulate cognitive activity and sustain engagement.

During phase 1 focus groups and questionnaires with carers, managers, relatives and people with dementia explored the acceptability of MARIO to people with dementia. Phase 2 gathered perceptions through focus groups with carers, managers, people with dementia and relatives of what they believed MARIO should be able to do in order to help people with dementia. In addition, researchers, with consent, also accessed the life history and personal interests of each person with dementia to inform the development of the MARIO applications. In phase three, the focus of this paper, an evaluation of MARIO was conducted. People with dementia were invited to engage with MARIO over a period of two months in each respective site and qualitative data were collected from people with dementia, carers, managers and relatives to ascertain their perceptions of the use and impact of MARIO. In addition in order to determine the respective costs and savings derived from using a social robot like MARIO, value maps for each of the different settings, namely hospitals, nursing homes, and communities were developed. However, these economic aspects are beyond the scope of this paper. This paper reports on the findings from the qualitative data collected in phase three, with the following aim.

### 2.4. Aim

To explore the perception and experiences of people with dementia and key stakeholders as regards the use and impact of the social robot, MARIO.

### 2.5. Methods

A qualitative interpretive descriptive design based on the work of Thorne [62] was used to explore the perceptions and experiences of people with dementia and key stakeholders as regards the use and impact of MARIO. Interpretive description is designed to give participants a voice about their own experiences. It is particularly appropriate when seeking to understand complex “phenomena”, such as those investigated in this study. Semi-structured one-to-one interviews using interview guides developed from the literature and directed by the research aims were used to collect the data. These guides were initially created in English and then subsequently translated by members of the Italian research team for use in Italy. Ethical approval was obtained in each of the three pilot sites (UK, Ireland, and Italy).

### 2.6. Sample

A purposive sample of 107 stakeholders (people with dementia, carers, relatives, managers) who were directly involved with MARIO across the three pilot sites participated. An overview of the number of participants involved at each site during phase 3; the MMSE range in each site, the number and duration of interactions with MARIO are presented in Table 2 below. A total of 195 engagements with MARIO were completed with people at different stages of dementia.

In the UK eight people living with dementia in their own homes (five females and three males), were involved. All had mild dementia (MMSE range 20–23) and six were 60+ years of age. Five relative/informal carers took part (three male and one female), and three were 60+ years of age. There were six managers (four female and two male), and most were in the 50+ age group. Two managers were responsible for managing dementia support groups, one was responsible for commissioning services for older people, and three were managers within the adult social care department.

In Italy, 20 people with dementia who were in-patients in a hospital (12 females and 8 males) participated most of whom were over 76 years of age and all had mild dementia (MMSE range 19–23). Eighteen relatives participated (13 female and five male), with the majority in the 70+ age group (*n* = 10). A total of 20 formal carers (13 females and 7 males) participated of these over 60% were geriatricians. Two managers were interviewed both of whom were male with an average age of 49 years.

Ten people with dementia living in long-term care participated in Ireland. Six had moderate dementia (MMSE range 14–19) two mild dementia (MMSE range 20–30) and two severe (MMSE range 3–13). Nine were over 70 years of age and one over fifty. Six female relatives aged between 40–59 years and eight formal carers, (6 females and 2 males), participated. Of the latter six were registered nurses and two were health care assistants. Five managers participated, (three female and two male) three were aged between 50–59 years and two. Numbers of participants in each pilot site, and number and duration of interactions are outlined in Table 1.

### 2.7. Data Analysis

All interviews were transcribed verbatim and directed qualitative content analysis based on the work of Hsieh and Shannon [63] was used to analyse the data. A coding framework for each stakeholder group data set was developed based on literature analysis and findings of prior research undertaken with people with dementia [64,65]. Four researchers (KM, TK, SW, EB) from the Irish pilot site were involved in the initial development of the data analysis coding frameworks. To ensure coding consistency, a sample of interview transcripts from each stakeholder group, (people with dementia, carers, managers and relatives) was then independently coded by researchers. Researchers worked independently but in pairs to analyse three transcripts from each respective stakeholder group. Then, inter-rater reliability testing was conducted on the set of codes they produced, using a Cohen’s ‘Kappa’ [66] that scientifically measures the degree of agreement between coders. The inter-rater reliability scores for each pair of researchers ranged from 0.67 to 0.76. Following this, a meeting was held to agree coding, examine any differences, resolve discrepancies, and agree the final coding frameworks. Then, the coding frameworks were discussed and shared with pilot partners in Italy and the UK who subsequently tested and confirmed the applicability of the frameworks to their respective data sets. The frameworks were then used to analyse the data across all pilot sites, G was the code for Ireland, S for the UK and I for Italy. The following codes were used: D is a person with dementia, R is a relative of the person with dementia, C is a carer of a person with dementia and M is a manager in the practice site. These participants are collectively referred to as stakeholders. The country and stakeholder codes used to report the findings are presented in Table 2.

## 3. Findings

Data analysis revealed five key themes: perceptions of MARIO, impact of MARIO, utilisation of MARIO applications and interfaces, challenges in the use of social robots in the real-world context of dementia care and improving MARIO.

### 3.1. Perceptions of MARIO

The findings revealed that most participants across all sites had positive perceptions of, and attitudes towards, MARIO and they were generally accepting of a social robot referring to the robot as ‘he’ or ‘she’, conceptualising him as an embodied presence or a ‘friend’.


*I can talk to her and she’s lovely and she’s tolerant. (GD1)*



*MARIO is like a friend. I really enjoyed this experience. (ID4)*


Some people with dementia in the community were initially wary about MARIO. However, as prolonged engagement occurred, the development of a mutual care relationship became evident in the way that the participants greeted and interacted with the robot. Sometimes asking questions about its well-being and telling it “Don’t worry pet” when they perceived there was something wrong with it or looking for the robot when it had left their company.


*MARIO and I have made a very close relationship over the last few months. (SD3)*



*Where did the man {MARIO} with the music go? (researcher tells her that MARIO had gone to the dayroom. She replies ‘Oh I would have gone with him if I had known (GD12).*


Participants with dementia reported that they liked that MARIO was non-critical and commented that it helped them forget they had dementia which in turn made them feel more confident, supported and they enjoyed the experience.


*She makes me feel normal. (GD13)*



*It has made me feel surer {confident} (ID10 Trial 2).*



*I look forward to using MARIO and I feel I am learning… (SD3).*


In addition, people with dementia across all sites expressed a desire to have a MARIO robot in their own home, intimating at the positive impact this might have on their lives.


*I enjoyed it, first time I went home I wanted one. (SD4)*



*Can you imagine if I could have one of these at home (ID3).*


Relatives across all the pilot sites were also mostly positive towards MARIO, seeing the robot as a source of interaction and entertainment as well as a companion and personal assistant that could help their relative with the challenges of living with dementia. Some commented that it took some time for people with dementia to get used to interacting with MARIO but that after spending time with the robot they became more confident users. Carers who were able to observe MARIO in action with their relative were able to give examples of MARIO’s impact and they understood what MARIO could do and were generally positive.


*Advantages would be companionship, reminders and having someone to talk to. You could have a conversation {with MARIO}. (SRC2)*



*I mean I don’t know but I think she’s getting companionship of a sort, she’s getting entertainment, diversion, fun with the conversations that the people with dementia directs or leads or you know, persuades out of her that are you know, the point, the touch point or the stuff on the screen. So it’s brilliant, it’s really good, love it, yeah its great (GR13).*



*Using MARIO in hospital, my father showed an improvement of his mood, anyway not only for this aspect MARIO can be useful in Hospital. It can improve the hospitalization of participants with dementia and reduce the risk of cognitive decline (IR4).*


Carers in residential care and carers/relatives in the community commented on both their own acceptance of MARIO and the acceptance by people with dementia. Some carers/relatives reported being initially sceptical about the value of MARIO. Overtime they changed their views after seeing MARIO in action and the impact the robot had on people with dementia. In particular, personalising the robot according to the needs of people with dementia went a long way towards changing carers views and them having a more positive disposition towards MARIO.


*They (participants) realised how much more useful it (MARIO) has become since it has been personalised (SRC3)*



*…think it is brilliant. It could really be helpful; mainly because you can personalise it (MARIO) (SRC4).*



*Since my mother forgets her medicines, MARIO helps her to remind her about daily medication. MARIO also notifies her about the hospital appointments…It is tailored to her needs. (IR9).*


However, within the hospital setting carers were overly optimistic about what MARIO would be able to do, some expected a fully independent robot and therefore were less positive at the end of the evaluation than they were at the start.


*My opinion about MARIO changed. Initially I believed that MARIO was able to do more things. Now I think that technology is not ready to give to participants a fully independent and operational companion robot. (IC3)*


In the context of residential care, the experience of working with MARIO did not really change the perceptions of people with dementia, carers, managers and relatives. For the most part, those who were positive from the outset remained so and those who were sceptical and believed that social robots had a very limited role in the context of people with dementia continued to do so.

### 3.2. Impact of MARIO

Across all three settings, all participants suggested that the main impact of MARIO for people with dementia was; increased cognitive engagement, autonomy, reduced loneliness, and isolation, all of which led to some improvement in their quality of life.

For most participants with dementia, moments of positivity were experienced and witnessed whilst they engaged with MARIO. Some carers and relatives described their surprise when participants in residential care, with quite advanced dementia, were able to concentrate and sustain engagement with MARIO, while using the applications, despite having a history of problems with attention span. Participants reported that MARIO had helped to focus the attention of the person with dementia engagement increased and they felt the person with dementia benefited from this engagement.


*…he could do it (use the painting app. on MARIO} … he spent 40 min one evening doing it which was great, 40 min like, even the nurses were surprised to see him doing it for 40 min (GR5).*



*I have seen the person with dementia attentive and engaged during their interactions with MARIO. They told me that interacting with MARIO was fun and pleasant, and I have seen their great enthusiasm… (IC9).*



*This is brilliant. Could get a lot out of it {MARIO}. (SD1)*


People with dementia were able to select from a menu of applications that were individualised to them. In the context of residential care, giving the person with dementia the opportunity to select what they wished to do, gave them choices that enhanced their autonomy. This was important to these participants as they reported that they sometimes felt bad about asking people for help.


*Asking them {care staff} things, like show me this and are you able to do that and I feel bad. (GD14)*


In the context of the community setting, most participants with dementia had milder levels of dementia and were living well with their condition. MARIO therefore had little impact on their autonomy or choice selection as they were generally able to already make autonomous choices.

In all three sites, people with dementia, carers and relative participants described the lives of people with dementia as routine, dominated with long periods of inactivity and little interaction with others. Participants with dementia reported that MARIO made them feel less lonely as the robot provided distraction, allowed for engagement with a wide variety of activities and facilitated interaction with family members. Across all settings, MARIO also provided a topic of conversation with family and carers as well as providing a conduit by which participants with dementia could connect with others. In addition, carers/relatives and managers commented on the multifaceted social activities which MARIO offered which they felt had the potential to reduce loneliness and enhance social engagement and interaction for people with dementia.


*MARIO could reduce and prevent the isolation and loneliness of the participants. (IM2)*



*Real potential to connect people with the community, more with family and friends. (SM2)*



*…she’d {person with dementia} have the different options of different things instead of just having the same thing—the television, playing bingo, the same...Just a couple of things that way because there would be more of an option with MARIO (GC18).*


Some participants with dementia reported that MARIO had improved their mood thereby improving their quality of life.


*He’d {MARIO} make you good... I always thought that he’d make you feel good (GD14).*



*It {MARIO} just cheers you up and makes me dead happy. (SD4)*


In addition, relatives in the residential care setting commented that it was the provision of extra activities for their relative that made a difference to daily living.

For most relatives in the residential and hospital setting, MARIO provided a diversion, something different, an embodied presence that provided companionship, connectivity and improved the overall mood of the person with dementia. Carers in residential care and carers/relatives in the community saw the personalisation of activities to the person with dementia as key to its positive impact.


*Once the data and everything else was collected, I was really impressed that it was individualised…to the actual client. That there was actual research done of their likes and dislikes and family background and everything else and yeah, good. (GC16)*



*It could really be helpful and always have done. Mainly because you can personalise this (SRC4).*


However, carers in all settings felt that the positive impact of MARIO was short lived and did not extend beyond the time of the interactions with MARIO. Therefore, they tended to describe the impact of MARIO as “in the moment only”, suggesting that more time and consistent use of MARIO was needed to assess the long-term significance of this type of intervention.

### 3.3. Utilisation of MARIO Applications and Interfaces

Voice recognition failed across all sites in circumstances where the background noise in the environment was too loud. The noisy environments in these real-world settings meant that MARIO frequently had difficulty processing what the person with dementia said. In addition, some participants with advanced stage of dementia had unclear speech, and patterns of speech that were atypical. This meant that participants in the residential care setting often needed to operate MARIO via the touchscreen either by hand or a stylus.

Across all sites, the two most popular applications were the *My Music* and *My Memories.* The *My Music* app tended to be the first option selected when engaging with MARIO. Most people with dementia were able to use the application independently and were observed to engage fully with it. They described it as enjoyable and commented on the positive impact it had on their mood.


*How did it make you feel when you listened to the music?*



*I felt good… (GD1).*



*I liked the music best, good music today...(SD2).*


Carers/relatives across all sites also commented positively on the impact of this app. In the residential setting people with dementia were observed dancing tapping their fingers or the floor with their foot to the music, singing along, and reminiscing about the content. In the hospital setting, carers commented on the benefits of the music app, as it prompted physical activity.


*She was dancing and singing…She was so excited when using MARIO (IC3)*


The second most popular app. across all sites was the *My Memory* reminiscence app. For people with dementia this app. facilitated their recall of happy memories. Likewise, relatives/carers and managers commented on this app’s importance in drawing on long-term preserved memories which stimulated the participant with dementia and created enjoyment for them.


*To look at the photos has made me remember the beautiful moments of my life (ID6).*



*So, I think looking at pictures and talking about them is—it’s good. (GR13)*



*The photos are really useful. (SRC1)*


### 3.4. Challenges to the Use of Social Robots in the Real-World Context of Dementia Care

Two main challenges emerged from the data: (i) negative attitudes/concerns towards the use of robots in care giving; (ii) the stage of dementia.

While most carers/relatives and managers were positive about MARIO, some expressed concerns regarding the future deployment of robots in dementia care. These concerns related to the fact that robots should not be a replacement for human interaction or carers.


*Mario must be perceived as an aid, not as a human being that will substitute the staff or the family. (IM2)*



*Note of caution that it doesn’t become a replacement for human interaction… (SM2)*



*…we used to have another fulltime occupational therapy assistant, once they retired, they weren’t replaced. So, I can’t see in any way that Mario would compensate in any way for the loss of that… (GC13).*


Instead some carers believed that social interactions needed to be with another human or even an animal in order to be beneficial. In addition, some did not believe that robots had the capacity to provide the care that they did, respond to cues or individualise their responses sufficiently to work effectively with people with dementia.

Carers, managers and relatives across all Sites also commented on the fact that the stage of dementia was an important consideration when deploying robots to work with people with dementia. They suggested robots were most useful at the mild to moderate stage of dementia because those with severe dementia may find it hard to understand the technology, use the touch screen or generally engage and interact without a lot of guidance and technical support.

### 3.5. Improving User Experience of MARIO

The key improvements suggested by most carers/relatives and managers across the pilot sites revolved around improving the speech recognition and adding monitoring and assessment devices for people with dementia to keep them healthy and safe.


*maybe a safety thing…If you could use Mario that way? …Like if it was in somebody’s home if they fell could they say ‘Mario, ring the ambulance’ or whatever? (GR16).*



*Could do more, support people to do more physical activity, tools to encourage more movement, how do you do this? Check someone is doing it? It would be really good. It would be brilliant, more mobile, build in exercises, help with medication, these are key elements (SM2).*


With regard to speech recognition, it was suggested by all participants that MARIO’s conversational ability needed to be developed further so that the robot could understand what people were saying, respond appropriately, and have more meaningful conversations. In addition, having a more humanoid type robot, with facial recognition, and with more autonomy were considered key to making MARIO more useful as a social robot for people with dementia. As regards the future of companion robots in dementia across all pilot sites, carer/relatives, managers and some people with dementia, believed that a MARIO type robot would be a useful addition and support. In particular, it was felt by some that MARIO would be suitable for people in the earlier stages of dementia and for those living in their own homes in the community. However, many of the participants with dementia in the community believed MARIO would probably be most useful for people more worse off than they were at that time.

Overall findings from this qualitative study demonstrate that the companion robot MARIO was an accepted part of social care for people with dementia and had an important role to play in combatting loneliness and increasing levels of engagement. The key strength of this project was that MARIO entered the real world of clinical practice for testing, development and evaluation in three different settings and countries. The applications were developed with regular feedback and testing by the potential end users, within the context in which they would eventually be deployed.

## 4. Discussion

The discussion focuses on four areas, acceptability, human-robot relations, social activities and social isolation and enhancing autonomy. These discussion themes are summarized below in Table 3.

### 4.1. Acceptability

There are divided views within the literature as to the acceptability of social robots in the care of people with dementia, with some researchers reporting that they are not acceptable and others that they are. Researchers who found that social robots are not acceptable report that this is because robots lack the capacity to perceive emotional cues or react appropriately [67,68,69,70] and that staff are concerned about sharing their working space with a robot [71,72]. Studies from the area of disability [68] and aged care [69] have reported similar issues. Ambivalent attitudes of staff towards robots, and in particular, the fear that robots would replace care staff, were uncovered in the MARIO study too. While MARIO was perceived as an important addition to older people’s daily routine, a minority of carers were not keen on having robots in practice areas and believed that robots did not have sufficient capacity to interpret and respond to the needs of people with dementia. Some carers and managers believed that any resources should be channelled towards increasing numbers of staff not buying robots and that robots should not be used to replace human carers.

Researchers who found that social robots are acceptable in the care of people with dementia report that robots can provide companionship, cognitive stimulation and reduce loneliness [9,47,48,49,50,56,73,74,75,76]. While these studies display promising results many were conducted over relatively short testing periods; two days [22,49,50,59,60], 2 weeks [77], 3 weeks [47] or 6 weeks [58], conducted in the participant’s home [47,48,50] or a simulated set up in a laboratory [59,60]. It is not known therefore if these findings would be replicated in the real world of practice and sustained over time. The MARIO findings that robots are acceptable to people with dementia are therefore important because they were conducted in the real world of practice, included the views of people with dementia and took place over 12 months and therefore strengthen the claims that social robots are acceptable in dementia care.

Researchers have also identified a number of factors that influence the acceptability of social robots including; perceived usefulness, trust, enjoyment, the opinions of the end user’s significant others and a robot platform that provides meaningful applications and places low technical demands [54,57,59,72,78,79,80]. The findings of MARIO suggest that the embodied presence of the robot is also important and that the personalisation of the applications to the user is correlated with increased engagement.

### 4.2. Human-Robot Relationships

The nature and desirability of human-robot relationships is also an area of divided opinion [81]. Some researchers argue that human-robot relationships are positive because robots can provide companionship [13,39,40,41,42] and time spent with a robot, because it is stimulating, can enhance communication between the person with dementia and other people [40,56,74,75,76]. Other researchers disagree arguing that developing a relationship with a robot is undesirable because it is dehumanising and unethical [67].

Researchers who report positive human-robot relationships have found that people with dementia often referred to the study robot as a friend [48,82,83]. This was a finding also in the current study as many people with dementia referred to MARIO as ‘he’ or ‘she’ or as ‘my friend’. In addition positive emotional responses have also been reported in studies involving the humanoid robot NAO [84] and PARO [73,84]. The benefits of small positive moments experienced throughout the day for people with dementia, such as those experienced during interactions with MARIO, should not be undervalued as it is believed that these significantly benefit the happiness, positive self-perception and overall quality of life of people with dementia [85]. Some researchers caution however that the robot-human relationship may not be sustained overtime as people with dementia lose interest in the robot [86,87]. However in contrast to these findings MARIO found evidence that the robot human relationship strengthened over time, although further studies of longer than three months are required to confirm this.

### 4.3. Social Activities and Social Isolation

Many studies conducted in long-term care and hospital care settings have found that participants’ lives were dominated by routine with long periods of inactivity, an absence of social participation, low levels of communication and high levels of loneliness [13,23,24,25,26,28,29,86,88]. Cook [89] suggests that “social death” can occur in residential care arising from a lack meaningful activity. This is especially so for people with dementia who have often experienced an on-going lack of stimulation and social interaction leading to lethargy, boredom, depression, social isolation, loneliness and poor quality care [30,31,89,90]. Some researchers argue that in this context, social robots should be considered as a way of increasing social activity, facilitating communication, reducing loneliness and providing opportunities for people with dementia to engage in meaningful activities [38,91]. Many researchers have found that interaction with a social robot can lead to more engagement with people, not only because the robot provides a topic of conversation, but also because engaging with the robot is cognitively stimulating [13,14,40,48,73,92,93,94]. Liang et al. [73] found that the social robot PARO had a positive impact on the communication between people with dementia and day centre care staff. This finding is supported by a number of other researchers [13,39,40,41,42] who also found that communication with staff and relatives improved following work with a social robot. Robinson et al. [57] found that work with robots that offered stimulation and entertainment led to increased levels of social engagement and increased the person with dementias ability to interact with other people. Chu et al. [17] found social robots provided sensory enrichment, social engagement and entertainment. They concluded that social robots can improve quality of life for people with dementia. Across all sites MARIO was found to facilitate conversations and social engagement providing participants with dementia the opportunity to converse with staff and relatives about their own life and that MARIO was effectively able to provide activities for people with dementia. While engaging with MARIO, participants with dementia spent less time alone and more time socially engaged and MARIO facilitated people with dementia to focus on their preferred activities for lengthy periods of time, even if they usually found it difficult to focus. MARIO provided a conduit for connection to family and friends and provided information on personal interests, giving the person with dementia the potential to engage more in conversations. Moyle et al. [58] explored whether social robots could promote social connectedness via video calls between relatives and participants with dementia who lived in long-term care. They also found that the robot increased opportunities to reduce social isolation and encouraged engagement.

### 4.4. Enhancing Autonomy

Many researchers have identified autonomy as a core attribute of the quality of care of older people [95,96,97] However, previous research has found that many older people living in long-term care have reduced levels of autonomy [95,96]. Researchers claim that giving older people the choice of what they want to do and allowing them to select personalised activities when working with a robot can enhance autonomy [17,57,87]. The MARIO findings support this claim as it was the autonomy given to people with dementia to make autonomous choices about what activities they wanted to do that was particularly valued.

## 5. Conclusions

Findings from this qualitative study demonstrate that the social robot MARIO was an accepted part of social care for people with dementia. The embodied presence of MARIO, the user-led design process and development of personalised activities led to a broad acceptance of the MARIO robot in dementia care amongst people with dementia, relatives, carers and managers. The findings confirm that social robots may have an important role to play in combatting loneliness, enhancing autonomy and increasing levels of engagement. With the current challenge of the global COVID-19 pandemic there are compelling reasons for long-term care facilities to utilise more social robots. Many long-term care facilities across the world have had to limit visitors, because of the pandemic, thereby reducing social contacts. Caleab-Solly [98] argues that telepresence robots could be used to help alleviate this social isolation. In addition, a call to action from the robotics community on the role of social robots in managing public health and infectious diseases appeared recently [99] with a specific call for increased adoption of social robots as the widespread quarantine of patients, is resulting in prolonged isolation of individuals from social interaction. Social robots such as the MARIO robot could be deployed to provide continued social activities, connection with friends and family and adherence to treatment regimens without fear of spreading disease. However concerns remain around the emotional capacity of robots. Future research should ensure that robot designs for use in dementia care possess more human-like features and enhanced capacity to communicate and understand the speech of people with dementia. In addition, the introduction of social robots needs to ensure that health care expectations are realistic and focus on promoting positive attitudes when preparing staff to work with the technology. Finally, future evaluation of the impact of social robots in dementia care needs to include longer testing and evaluation periods with larger sample sizes. Despite the limitations, promising trends as to the positive impact of MARIO on improving social and cognitive health and the ability to reduce loneliness is evident in the context of using a companion robot such as MARIO for older people with dementia.

## 6. Limitations

Given the absence of a fully autonomous robot and the constant presence of the researcher it is difficult to come to categorical conclusions regarding the impact of MARIO. Further studies with larger sample sizes than the one used in MARIO and longer duration are required.

## Figures and Tables

**Figure 1 ijerph-17-08621-f001:**
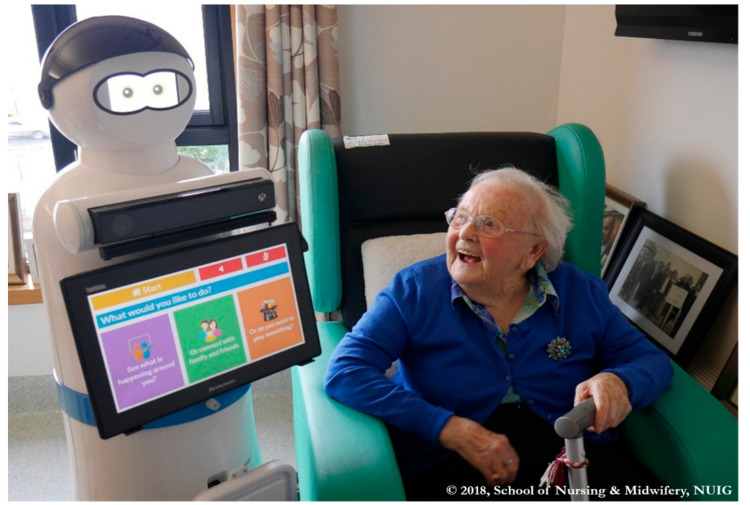
A resident of the nursing home in Ireland interacting with the MARIO social robot as part of her daily routine. (The first author has copyright of this figure).

**Table 1 ijerph-17-08621-t001:** Participants numbers. Interactions with MARIO.

	Hospital (Italy)	Long-Term Care (Ireland)	Community (UK)	Across Sites Total
Participant Categories				
People with dementia	20	10	8	38
Relatives/Carers	18	6	4	28
Formal Carers	20	8	0	28
Managers	2	5	6	13
Total Participants across categories	60	29	18	107
*Number of Interactions with MARIO*	Hospital	Long-Term Care	Community	Across Sites
Number of interactions with MARIO.	75	96 ^1^	24	195
*Duration per interaction*	Hospital	Long-Term Care	Community	Across sites
Average with MARIO per session.	43.7	35	60	41.3
TOTAL INTERACTIONS				Values
Total duration of interactions with MARIO (minutes)— Mean ± SD Range				198.62 ± 101.09 15—524
Number of Interactions between people with dementia and MARIO— Mean ± SD Range				5.13 ± 3.44 1—12

^1^ In the residential care setting 3 participants completed one, seven, and four MARIO engagements respectively, whereas all the other 7 residents completed twelve engagements.

**Table 2 ijerph-17-08621-t002:** Pilot site stakeholder codes.

Stakeholders	Ireland: Residential Care Setting	UK: Community Setting	Italy: Hospital Setting
Person with Dementia	GD	SD	ID
Relative	GR	SRC *	IR
Carer	GC		IC
Manager	GM	SM	IM

* Indicates that some relatives fulfilled the role of carer in the community setting.

**Table 3 ijerph-17-08621-t003:** Summary of Discussion.

Discussion Themes	Literature	MARIO
Acceptability	There are divided views within the literature as to the acceptability of social robots in the care of people with dementia, with some researchers reporting that they are not acceptable and others that they are.	Robots were found to be acceptable to people with dementia. In addition the embodied presence of the robot and personalisation of the applications to the user was correlated with increased engagement.
Human-Robot Relationships	The nature and desirability of human-robot relationships is an area of divided opinion.	MARIO found evidence that the robot human relationship strengthened over time, many people with dementia referred to MARIO as ‘he’ or ‘she’ or as ‘my friend’.
Social Activities and Social Isolation	Many studies conducted in long-term care and hospital care settings have found that participants’ lives were dominated by routine with long periods of inactivity, an absence of social participation, low levels of communication and high levels of loneliness.	MARIO provided a conduit for connection to family and friends and provided information on personal interests, giving the person with dementia the potential to engage more in conversations.
Enhancing Autonomy	Many researchers have identified autonomy as a core attribute of the quality of care of older people	The MARIO findings support this claim as it was the autonomy given to people with dementia to make autonomous choices about what activities they wanted to do that was particularly valued.

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
