# Peer review of "The Perceptions of People with Dementia and Key Stakeholders Regarding the Use and Impact of the Social Robot MARIO"

_ijerph, 2020, doi:10.3390/ijerph17228621_

Round 1
Reviewer 1 Report
Using robot technology to improve benefits for people with dementia is a relevant issue that should be continued in future studies that exceed the sample size and the time of the intervention.
Author Response
Reviewer 1
|
REVIEWER1 |
|
|
|
Using robot technology to improve benefits for people with dementia is a relevant issue that should be continued in future studies that exceed the sample size and the time of the intervention |
Thank you for your review and the time taken to help improved the paper. We have added the sentence in the imitations section in order to address your comments. Thankyou |
Further studies with larger sample sizes than the one used in MARIO and longer duration are required.
|
Reviewer 2 Report
Thanks you for this fascinating report. This is an area of potential great significance and it is very interesting to read an evaluation to a trial of robotics applied to care. Your work is very clearly written and of sufficient detail and I do appreciate that in a paper of this kind there has to be a trade-off between description and substantive content. On the whole I think you have the balance right but it does leave an itch for a more theorised account - you don't really touch on some of the more substantive theoretical issues for example the ontological and moral status of the robot, whether the 'relationship' between the PWD and the robot is in a sense deceptive (one of bad faith). I am a little sceptical about whether it is necessary to package the functionality of the robot in a humanoid form. Might the dame ends be achieved by a device that doesn't attempt to mimic the human or is that an essential part of its effect? If it is then what implications might this have for human carers (professional and lay)? There is so much nursing theory for example that emphasises the unique role of care but can a robot care? I understand that you would have to write a very different paper to address these points.
I think that what you could do in this paper is say a little bit more in the introduction about the recent history and evolution of care robots as well as saying something a bit more specific about this example. I presume that all it's functions were linguistic/ visual as opposed to doing physical tasks?
I also think you could say a little more about your participants for example regarding mental capacity. You do say all gave consent so I presume that all participants had capacity and never lost capacity, but did engaging with a robot challenge capacity in any way (I am guessing that most PWD have never encountered a robot like this before?)? In the UK there are specific laws about mental capacity so perhaps a comment or reference to how this was negotiated across the various sites.
This is a nice paper and well-written but it has left this reader asking a lot more questions.
Author Response
Reviewer 2.
|
Issue |
Response |
Change Made |
|
REVIEWER 2 |
|
|
|
you don't really touch on some of the more substantive theoretical issues for example the ontological and moral status of the robot, whether the 'relationship' between the PWD and the robot is in a sense deceptive (one of bad faith).
|
Thank you for the comments and the time taken to help improve this paper. We appreciate that robots need to be deployed with full awareness of ethical and moral issues in order to prioritise the wellbeing of people with dementia. We therefore added this sentence to the ethical section.
The issue of deception was addressed in another publication by one of the authors, (Whelan, 2018). In this comprehensive literature review no empirical evidence was found to support the notion that people with dementia are deceived through their relationship and usage of social robots.
Of course, the possibility of deception occurring should be monitored in the future as the capability of robots continues to develop, but, given the already published paper we feel that a full discussion of these important ethical and moral issues is beyond the scope of this paper. |
p. 4 ‘The study was guided by experts in ethics who developed and implemented an ethical framework’. |
|
I am a little sceptical about whether it is necessary to package the functionality of the robot in a humanoid form.
|
We have changed the wording to address this comment. We hope this comment is helpful. An extensive review of the literature that investigated the factors that impact the acceptability of robots for people with dementia (Whelan, 2018) found that robots were more acceptable in a dementia context if the robots had humanlike facial features. It also found that people with dementia were more likely to engage with the robot, and through it applications that supported their psychosocial functioning, if the robot was embodied and sharing their physical space (Tapus et al., 2009). People with dementia are also known to engage longer with stimuli that possess social attributes than stimuli that are unsociable (Cohen-Mansfield et al., 2010).
|
p. 2 ‘include having humanlike facial features, being an embodied presence’
|
|
There is so much nursing theory for example that emphasises the unique role of care but can a robot care? I understand that you would have to write a very different paper to address these points |
We agree with the reviewer that a detailed discussion about the unique capacity of humans to provide care is beyond the scope of this paper. However, we have amended the paper to stress that the aim of MARIO is to support the psychosocial wellbeing of the user through providing cognitive stimulation, engagement in person-centered meaningful stimulating activities. These activities supplement rather than replace the care given by human caregivers. We recognise, the primary importance of human to human contact in care and that humans do have the capacity to provide deep care, whereas robots do not (Coeckelbergh, 2010, p.183). However, there is evidence that robots may have the potential to contribute, enhancing existing care provision. Robots, including MARIO, who has no functional capability to address physical needs, are designed to be an addition to human care givers (Moon et al., 2012; Roy et al., 2000; Mast et al., 2010) or be a resource for users within the home (Sorell & Draper, 2014). |
p. 3 ‘MARIO is a social robot whose functions aim to support the psychosocial wellbeing of people with dementia, through supplementing the care provided by human carers. MARIO has no functional capacity to address a person’s physical needs’. |
|
I think that what you could do in this paper is say a little bit more in the introduction about the recent history and evolution of care robots as well as saying something a bit more specific about this example. I presume that all it's functions were linguistic/ visual as opposed to doing physical tasks?
|
Thank you, we have added a new piece and we have made clear that MARIO does not have capacity to undertake any physical tasks. |
p. 3 ‘MARIO is a social robot whose functions aim to support the psychosocial wellbeing of people with dementia, through supplementing the care provided by human carers. MARIO has no functional capacity to address a person’s physical needs’. |
|
I also think you could say a little more about your participants for example regarding mental capacity. You do say all gave consent so I presume that all participants had capacity and never lost capacity, but did engaging with a robot challenge capacity in any way (I am guessing that most PWD have never encountered a robot like this before?)? In the UK there are specific laws about mental capacity so perhaps a comment or reference to how this was negotiated across the various sites. |
The participants in the UK and Italy all had mild dementia with MMSE range 19–23 and had capacity to consent. In Italy capacity was determined by a psychologist experienced in dementia research and in the UK researchers adhered to all legal requirements, with qualified health professionals who adhered to ethical standards and practices. In Ireland participants had mild to severe dementia. Here the consent process and assessment for capacity for consent adhered to the Irish Assisted Decision Making Act (2015) and the regulatory guidelines from the national consent policy (HSE 2017). This meant that the researcher who was a qualified nurse took considerable time using person-centred approaches to establish rapport and knowledge of the individual participants in order to establish capacity for consent. This work involved communicating with caregivers who knew the individuals well and used established practices for establishing the capacity and consent of people with dementia (Murphy, 2015) that are in accordance with the guidelines of the British Psychological Society (Herbert, 2019; Dobson, 2008). All three site used the principles of process consent (Dewing, 2007) whereby ongoing consent/assent was assessed and monitored throughout the research. The participants’ capacity was not affected by the robot. They had not encountered a robot before, were curious about MARIO and wanted to be involved in the study.
|
Page 4.The ethics section now reads All participants, including people with dementia gave their informed consent for inclusion before they participated in the study and confidentiality was maintained. Suitably qualified health professionals or psychologists, at each pilot site, ensured that participants had capacity to consent following procedures that conformed to national laws, regulations, and best practice in dementia research. Process consent was utilised, in that, consent was sought, not just for involvement in the overall research but, consent was checked again for each interaction with MARIO. The study was guided by experts in ethics who developed and implemented an ethical framework, and the study was conducted in accordance with the Declaration of Helsinki, and the study was approved by Research Ethics Committees in Ireland (REC, NUI, Galway) UK, (REC, Stockport Metropolitan Council) and Italy (REC:Casa Sollievo della Sofferenza) In addition, it was recognised that there was a need for careful management of the disengagement process between MARIO and person with dementia, particularly for those who had spent longer times with MARIO. Issues connected with the disengagement process were identified as potential ethical challenges and a disengagement plan was utilised’.
|
Reviewer 3 Report
The robot described in the present manuscript can be an important help for people with dementia as the authors explain.
In the health literature, currently there are not works concerning the use of robots in healthcare, even though professionals will have to be involved with them in the near future.
The manuscript is easy to follow and makes possible to understand the possibilities and improvements needed.
Some things to improve:
- Figure 1 has no capitation
- Material and methods. I think that more information about the background of MARIO, should be added, for example who did the design and produced it.
- Even though that in the discussion the authors consider different aspects in order to have a summary of them, a table with the main issues could be added.
- Have the authors taken into account the costs/savings derived of using a robot in this setting?
Author Response
|
Reviewer 3 |
|
|
|
|
Thankyou for your comments and the time taken to help improve the paper. We have reponded to and addressed where we can your comments |
|
|
· Figure 1 has no capitation
|
Thank you now included |
Has been added |
|
· Material and methods. I think that more information about the background of MARIO, should be added, for example who did the design and produced it |
New piece added |
P 3 ‘A multidisciplinary trans-European consortium of researchers, clinical practitioners from community, hospital and residential care settings, ICT specialists and industrial partners with expertise in robotics were assembled with the aim of developing a social robot. In total, the MARIO consortium brought together the skills and expertise of 10 partners from 6 countries. MARIO is a social robot whose functions aim to support the psychosocial wellbeing of people with dementia, through supplementing the care provided by human carers. MARIO has no functional capacity to address a person’s physical needs’ |
|
· Even though that in the discussion the authors consider different aspects in order to have a summary of them, a table with the main issues could be added.
|
A summary table has been added outlining the themes. |
Table 4 added |
|
· Have the authors taken into account the costs/savings derived of using a robot in this setting?
|
The economic aspects were beyond the scope oft he paper however, we have added a small piece identifying what we did to keep account of costs |
P 5 ‘In addition order to determine the respective costs and savings derived from using a social robot like MARIO, value maps for each of the different settings, namely hospitals, nursing homes, communities were developed. However, these economic aspects are beyond the scope of this paper.’ |